# TFRC Ablation Induces Insufficient Cartilage Development Through Mitochondrial p53 Translocation-Mediated Ferroptosis

**DOI:** 10.3390/ijms26062724

**Published:** 2025-03-18

**Authors:** Yidi Wang, Xi Wen, Yutong Guo, Yixiang Wang, Yan Gu

**Affiliations:** 1Department of Orthodontics, Peking University School and Hospital of Stomatology & National Center for Stomatology & National Clinical Research Center for Oral Diseases & National Engineering Research Center of Oral Biomaterials and Digital Medical Devices & Beijing Key Laboratory of Digital Stomatology & NHC Key Laboratory of Digital Stomatology & NMPA Key Laboratory for Dental Materials, No. 22, Zhongguancun Avenue South, Haidian District, Beijing 100081, China; 18915891068@163.com (Y.W.); pkuwenxi@163.com (X.W.); guoyutong6@126.com (Y.G.); 2Central Laboratory, Peking University School and Hospital of Stomatology, No. 22, Zhongguancun Avenue South, Haidian District, Beijing 100081, China

**Keywords:** transferrin receptor 1 (TFRC), SLC39A14, ferroptosis, chondrogenesis, cartilage development, mitochondrial p53 translocation

## Abstract

The mandibular condyle cartilage serves as a principal zone for mandible growth, and any dysplasia could contribute to skeletal mandibular hypoplasia (SMH). The aim of the study was to further explore how TFRC signaling regulates condylar cartilage development. In this study, TFRC, SLC39A14, chondrogenic markers and ferroptosis-related signals were detected in the condylar cartilage of postnatal mice and *Tfrc* cartilage conditional knockout (*Tfrc*-cKO) mice at different time points through immunofluorescence, immunohistochemical staining and qPCR assays. The overexpression and knockdown of TFRC in the ATDC5 cell line were used to investigate its role in a specific biological process. Co-immunoprecipitation was used to verify protein–protein interaction in vitro. Ferroptosis inhibitor Fer1, Ac-Met-OH and DFP were used for an in vitro rescue assay. The temporomandibular joint injection of DFP was used to rescue the cartilage phenotype in vivo. Our results verified that TFRC was crucial for condylar cartilage development. TFRC ablation led to condylar cartilage thickness and condyle length alterations and induced the ferroptosis of chondrocyte by upregulating SLC39A14. Mitochondrial p53 translocation was involved in the TFRC–SLC39A14 switch by SLC39A14 ubiquitination degradation. Fer1, Ac-Met-OH and DFP inhibited ferroptosis and restored chondrogenic differentiation in vivo. The temporomandibular joint injection of DFP could rescue the cartilage phenotype. In summary, this study reveals that TFRC influences postnatal condylar cartilage development through mitochondrial p53 translocation-mediated ferroptosis, which provides insights into the etiology, pathogenesis, and therapy of mandibular hypoplasia and even systemic articular cartilage dysplasia.

## 1. Introduction

The mandibular condyle cartilage serves as a principal zone for mandible growth, influencing both the mandible length and height [1]. Postnatally, the condylar cartilage progressively evolves into four distinct layers including the surface layer, proliferative layer, hypertrophic layer and calcified cartilaginous layer. During condylar growth, the chondrocytes undergo direct trans-differentiation into bone cells, a process documented in the previous studies [2,3]. Insufficient cartilage development is a determinant factor in skeletal mandibular hypoplasia (SMH) [4,5], a common deformity in hard tissue development that often leads to an aesthetically displeasing maxillofacial appearance [6].

In some cases, SMH patients also endure the distressing symptoms of obstructive sleep apnea syndrome (OSAS), which affects 2–4% of children and significantly impairs their quality of life [7,8]. Functional appliances, such as Twin-block and activator appliance, are commonly used for non-surgical growth modification in SMH patients during the peak of mandibular growth and development [9,10]. However, the efficacy and stability of functional treatment remain uncertain [9,10,11]. Therefore, it is imperative to investigate the pathogenesis of mandibular condyle deficiency and explore adjuvant therapies to foster mandibular growth.

Transferrin receptor 1 (TFRC) is expressed in a wide range of tissues and cells, especially those undergoing rapid proliferation or requiring an urgent energy supply [12]. Transferrin (TF), which has an abundance of ferric ions (Fe^3+^), binds to TFRC located on the cell membrane and forms an endosome via cellular endocytosis. In an acidic environment, ferric ions (Fe^3+^) are reduced by the STEAP3 protein to ferrous ions (Fe^2+^), which are then released through the Fe^2+^ membrane transport protein DMT1 to meet cellular physiological demands. Following this process, TFRC is recycled back to the cell membrane with the endosome [13,14]. Our previous research suggests that the TF–TFRC signaling pathway plays a crucial role in mandibular growth, and TF could potentially serve as a serological biomarker for predicting condyle growth during puberty [15]. The deletion of *Tfrc* in neural crest cells causes micrognathia in mice [16]. Considering condyle cartilage dysplasia results in SMH, we chose TFRC as our target. To date, the underlying mechanism by which TFRC determines condylar growth remains elusive.

Ferroptosis is initiated by an increase in catalytically active iron ions within the cell, forming a labile iron pool [17,18]. TFRC is responsible for transporting transferrin-bound iron (TBI) ions. Recently, SLC39A14, a non-transferrin-bound iron (NTBI) candidate transporter has garnered significant attention, owing to its effect on the labile iron pool [19]. Besides abnormal iron metabolism in cells, ferroptosis is characterized by reactive oxygen species (ROS) generation and lipid peroxidation [20,21]. In addition, during the process of ferroptosis, prostaglandin-endoperoxide synthase 2 (PTGS2), NADPH oxidase 1 (NOX1), acylcoenzyme A synthetase long chain family member 4 (ASCL4) and nuclear receptor coactivator 4 (NCOA4) were upregulated, while glutathione peroxidase 4 (GPX4), solute carrier family 7 member 11 (SLC7A11) and ferritin heavy chain 1 (FTH1) were downregulated [22,23,24,25,26].

Considering that maintaining the dynamic balance of iron ion levels through TFRC and SLC39A14 is essential for life activities and that TFRC may regulate condylar cartilage development and mandibular growth, it is necessary to explore whether TFRC and SLC39A14 affect chondrogenic differentiation through ferroptosis and understand the underlying mechanisms involved.

In this study, we aimed to explore the role of TFRC in condylar cartilage development and elucidate the mechanism of TFRC–SLC39A14 switch in cartilage hypoplasia. Here, we demonstrate that the stable expression of TFRC is crucial for the fate of the articular cartilage and that the upregulation of SLC39A14 with excess non-transferrin-bound iron (NTBI) transportation would induce ferroptosis. The dominant expression switch from TFRC to SLC39A14 in chondrocytes is mediated by mitochondrial p53 translocation-induced SLC39A14 ubiquitination and degradation. Our findings have deepened the understanding of the roles played by TFRC and SLC39A14 in cartilage development, as well as the regulatory interplay between TFRC and SLC39A14, offering new perspectives for the etiology, pathogenesis and therapy of SMH.

## 2. Results

### 2.1. The Expression of sTFRC Decreases in SMH Patients at CVM Stage 3

The patients with SMH exhibited a protrusive profile, small jaw and deep overjet (Figure 1A,B). To further investigate iron metabolism during the pubertal growth peak, we analyzed the serum samples of 27 normal subjects at CS1–6 for various iron metabolism indicators in addition to the previously detected serum TF level [27]. Notably, the serum soluble transferrin receptor 1 (sTFRC) level reached its peak value at CS3 compared to CS6. Serum iron and ferritin levels remained stable during CS1–6 (Figure 1C). Detailed information regarding these findings is provided in Appendix A Table A1. Additionally, serum samples were also collected and compared from four SMH patients and four normal subjects at CS3, revealing that the sTFRC level of the SMH patients was significantly lower than that of the normal subjects (Figure 1D). Further details are available in Appendix A Table A2.

### 2.2. The Relationship Among Condylar Cartilage Development and the Expression of TFRC and SL39A14

Due to the challenges in obtaining samples of adolescent condyle tissue, we tested the relationship between condylar cartilage development and the expression of TFRC and SLC39A14 in mice at different time points. The results showed that the mandibular condyles of newborn mice were predominantly composed of transparent cartilage (0 week). As the mice grew, the condyle gradually elongated and ossified, with cartilage-covered parts forming, particularly during the newborn to postnatal 4-week period (Figure 2A). Safranin O/fast green staining indicated that the condyle of newborn mice was mainly composed of cartilage, which became relatively thin over time, with mature hypertrophic cartilage being replaced by bone trabecula (Figure 2B). Considering the crucial roles of TFRC and SLC39A14 in maintaining iron ion levels, we analyzed the expressions of TFRC, SLC39A14 and the chondrogenic biomarker ACAN in the mouse condyle tissues. IF demonstrated that TFRC expression was mainly concentrated in the cartilage proliferative layer, hypertrophic layer and subchondral bone of the condyle after 4 weeks. IHC also revealed a slight distribution of SLC39A14 in the proliferative and hypertrophic layers of condylar cartilage after birth, with a slight decrease in expression at 8 weeks. Furthermore, IHC showed that ACAN expression was abundant in the proliferative and hypertrophic layers during condyle growth progression (Figure 2B). The semi-quantitative calculation results of IF and IHC are shown in Figure 2C. Total iron content maintained at a degree in condylar cartilage from 0 w to 12 w mice (Figure 2D). qPCR analysis showed that the expression of *Tfrc*, *Sox9* and *Acan* were significantly increased and peaked while the expression of *Slc39a14* reached its low value at 8 weeks postnatal (Figure 2E).

### 2.3. Generation of Chondrogenic Conditional Knockout Mice with Weaker Condylar Chondrogenic Differentiation Ability

To further determine the relationship between TFRC and SLC39A14, as well as their roles in condylar chondrogenic development, Col2-Cre^ERT^; *Tfrc*^fl/fl^ mice were successfully generated and genotyped using PCR (Figure A1A,B). The length of the body, excluding the tail, of the *Tfrc*-cKO mice (6.17 ± 0.21 cm) was significantly shorter than that of the control mice (8.37 ± 0.15 cm). The mandibular length in the *Tfrc*-cKO group was 10.32 ± 0.34 mm, while the length in the control group was 13.60 ± 0.08 mm. The condylar length of cKO mice was 2.38 ± 0.03 mm and that of the control group was 3.8 ± 0.17 mm (Figure 3A). The condylar cartilage in the control mice was notably thicker (Figure 3B). Safranin O/fast green staining showed that the condylar cartilage was thinner, especially the condylar proliferative layer containing typical semi-circular dividing chondrocytes (arrow pointed) in the *Tfrc*-cKO mice (Figure 3C). The semi-quantitative calculation results of Safranin O (%) are shown in Figure 3D.

The expression of the chondrogenic differentiation markers, *Sox9* and *Acan,* in the condylar cartilage at the mRNA level were decreased in the *Tfrc*-cKO mice group compared to the control group. The Western blot also showed that SOX9 and ACAN were decreased in the *Tfrc*-cKO mice. These demonstrated there was weaker chondrogenic differentiation in the condylar growth of the *Tfrc*-cKO mice (Figure 3E,F).

### 2.4. Negative Correlation Expression Between TFRC and SLC39A14 in Condylar Cartilage of Tfrc-cKO Mice

Considering that TFRC and SLC39A14 are TBI- and NTBI-ion transporters, respectively, and the accumulation of iron ions can cause ferroptosis, we detected TFRC and SLC39A14 expression in the condylar cartilage of the mice. IHC and IF displayed that the expression of TFRC was knocked out and that SLC39A14 was increased in the proliferative and hypertrophic layers of the condyle in the *Tfrc*-cKO mice (Figure 4A). The semi-quantitative calculation results of IF and IHC are presented in Figure 4B. There is more iron content shown in the iron assay but less TF expression shown in the IHC results within the cartilage tissue of the *Tfrc*-cKO mice (Figure 4A,C). This demonstrates that NTBI was accumulated in the cartilage of the *Tfrc*-cKO mice.

TFRC-related ferroptosis biomarkers were examined in the condylar cartilage of the control and *Tfrc*-cKO mice. The Western blot results showed TFRC, TF and GPX4 were downregulated while SLC39A14, NOX1 and PTGS2 were upregulated at protein level in *Tfrc*-cKO mice (Figure 4D). qPCR results showed that the expression of *Tfrc* and *Gpx4* was lower while the expression of *Slc39a14*, *Ptgs2*, *Ascl4* and *Ncoa4* was higher in the cartilage tissue of the *Tfrc*-cKO mice at mRNA level (Figure 4E). The expression trend of TFRC, SLC39A14, PTGS2, NOX1 and GPX4 was consistent with the qPCR results.

### 2.5. TFRC Promotes Chondrogenic Differentiation of ATDC5 Cells Through Negatively Regulating SLC39A14 Expression

To investigate the effect of TFRC on the chondrogenic differentiation of the ATDC5 cell line, *Tfrc* was knocked down by a specific siRNA. qPCR analysis confirmed that the *Tfrc* mRNA level decreased while that of *Slc39a14* increased in the *siTfrc* group (Figure 5A). Western blot analysis showed that TFRC and TF at protein level decreased in the *siTfrc* group. The expression of SLC39A14 increased at protein level in the *siTfrc* group (Figure 5B). Both the qPCR and Western blot analysis demonstrate that the expression of chondrogenic markers SOX9 and ACAN decreased in the *siTfrc* group (Figure 5A,B). Alcian blue staining showed weakened chondrogenic activity in the micromass cultivation from the *siTfrc* group (Figure 5C).

For additional confirmation, we transfected *Tfrc* overexpression plasmids into the ATDC5 cell line. TFRC and TF expression were notably elevated in the group of ATDC5 cells transfected with the *Tfrc* overexpression plasmid (*OE-Tfrc)* compared to the group transfected with an empty vector (*OE-EV*). The results showed that SLC39A14 was downregulated while the chondrogenic markers SOX9 and ACAN were upregulated at both the mRNA and protein levels in the *OE-Tfrc* group (Figure 5D,E). Alcain blue staining indicated more robust chondrogenic differentiation through the micromass cultivation in the *OE-Tfrc* group (Figure 5F).

Then, we validated the negative regulatory relationship between *Tfrc* and *Slc39a14* in chondrogenic differentiation by the knockdown of both *Tfrc* and *Slc39a14*. In the *siTfrc* + *siSlc39a14* group, both TFRC and SLC39A14 expression was knocked down. The expression of the chondrogenic markers ACAN and SOX9 at the mRNA and protein levels were elevated in the *siTfrc* + *siSlc39a14* group (Figure A2A,B). Alcain blue staining of the micromass cultivation indicated more chondrogenic differentiation in the *siTfrc + siSlc39a14* group (Figure A2C). These data indicate that TFRC promotes the chondrogenic differentiation of ATDC5 cells through negatively regulating SLC39A14 expression.

### 2.6. TFRC Promotes Chondrogenic Differentiation of ATDC5 Cells Through Regulating Ferroptosis

To delve deeper into the mechanism underlying the chondrogenic differentiation influenced by the TFRC–SLC39A14 expression switch, we examined the ferroptosis-related biomarkers in vitro during the chondrogenic induction of ATDC5 cells in a low-serum medium. TFRC and TF were downregulated, the iron level was elevated, and the glutathione level was decreased in the *siTfrc* group (Figure 6A). Confocal microscopy images of the MitoFerroGreen staining demonstrated increased iron accumulation in the mitochondria of the *siTfrc* group (Figure 6B). Flow cytometry of the ROS revealed that the total percentage of DCFH-DA^+^ cells in the *siTfrc* group was 45.35%, which was higher than 40.14% in the *siNC* group. A C11-BODITY fluorescence probe was used to detect lipid peroxidation. The results showed that the ratio of oxidation to reduction was higher in the *siTfrc* group (37.11% vs. 60.53%) compared to the *siNC* group (27.46% vs. 71.74%). These results indicate that there was increased lipid peroxidation in the *siTfrc* group (Figure 6C). qPCR showed that the expression of *Ptgs2*, *Nox1*, *Ascl4* and *Ncoa4* increased while that of *Gpx4* and *Fth1* decreased in the *siTfrc* group (Figure 6D). Western blot analysis demonstrated that the expression levels SLC7A11, GPX4 and FTH1 were reduced while those of PTGS2, NOX1 and NCOA4 were elevated in the *siTfrc* group (Figure 6E). These results indicated that *Tfrc* knockdown alleviates the chondrogenic differentiation of ATDC5 cells by promoting ferroptosis.

On the contrary, *OE-Tfrc* exhibited decreased intracellular iron content and increased glutathione levels in the ATDC5 cells (Figure 6F). Confocal microscopy images of MitoFerroGreen staining showed that there was lower iron accumulation in the mitochondria of the *OE-Tfrc* group (Figure 6G). Flow cytometry analysis showed a decrease in ROS generation, with 40.53% DCFH-DA^+^ cells in the *OE-Tfrc* group compared to 44.62% in the *OE-EV* group. The redox ratio was more balanced in the *OE-Tfrc* group (33.16% reduced vs. 65.74% oxidized) compared to the *OE-EV* group (27.48% reduced vs. 71.30% oxidized) (Figure 6H). At the mRNA level, the expression of the ferroptosis-related genes, *Ptgs2*, *Nox1* and *Ncoa4,* was maintained or decreased while that of *Gpx4, Slc7a11* and *Fth1* increased in the *OE-Tfrc* group (Figure 6I). Western blot analysis confirmed these findings, showing maintained or decreased expressions of PTGS2, NOX1 and NCOA4, as well as elevated expressions of GPX4, SLC7A11 and FTH1 in the *OE-Tfrc* group at protein level (Figure 6J). Collectively, these results suggest that *Tfrc* overexpression promotes chondrogenic differentiation and alleviates ferroptosis in ATDC5 cells.

Additionally, the regulatory relationship of *Tfrc* and *Slc39a14* in ferroptosis was verified by the specific double knockdown of siRNA. There was decreased intracellular iron content and increased glutathione levels in the *siTfrc* + *siSlc39a14* group (Figure A3A). MitoFerroGreen staining showed that there was lower iron accumulation in the mitochondria of the *siTfrc* + *siSlc39a14* group (Figure A3B). Flow cytometry analysis showed a decrease in ROS generation, with 74.69% DCFH-DA^+^ cells in the *siTfrc* + *siSlc39a14* group compared to 85.25% in the *siTfrc* group. There was less lipid peroxidation in the *siTfrc* + *siSlc39a14* group (46.06% reduced vs. 50.66% oxidized) compared to the *siTfrc* group (20.85% reduced vs. 75.27% oxidized) (Figure A3C). There was an increased expression of the chondrogenic marker anti-ferroptosis factor *Gpx4* and a decreased expression of the ferroptosis-related markers *Ptgs2*, *Ncoa4* and *Ascl4* in the *siTfrc* + *siSlc39a14* group (Figure A3D). The specific knockdown of *Tfrc* and *Slc39a14* could promote the expression, at protein level, of GPX4 and inhibit that of PTGS2, NOX1 and NCOA4 (Figure A3E). The findings suggest that an imbalance between the expressions of TFRC and SLC39A14 triggers ferroptosis, leading to the disruption of normal chondrogenic differentiation in chondrocytes.

### 2.7. SLC39A14 Knockdown Promotes Chondrogenic Differentiation of ATDC5 Cells by Inhibiting Ferroptosis

To investigate the interplay between TFRC and SLC39A14 and the role of SLC39A14 on chondrogenic differentiation, we knocked down SLC39A14 expression by siRNA. qPCR and Western blot analysis showed decreased SLC39A14 expression but increased TFRC expression in the *siSlc39a14* group (Figure A4A,B). There was also increased SOX9 and ACAN expression at the mRNA and protein levels in the *siSlc39a14* group (Figure A4A,B). Alcian blue staining showed that chondrogenic differentiation was more active in the *siSlc39a14* group (Figure A4C). qPCR and Western blot showed that there was increased GPX4 expression and decreased PTGS2, NOX1 and NCOA4 expression in the *siSlc39a14* group (Figure A4D). These suggest that *Slc39a14* ablation could upregulate *Tfrc* expression, thereby promoting chondrogenic differentiation by inhibiting ferroptosis.

### 2.8. The Switch of TFRC–SLC39A14 Expression Regulates Ferroptosis Through Mitochondrial p53 Translocation in ATDC5 Cells

To investigate the underlying mechanism of the TFRC–SLC39A14 expression switch, a String network analysis (https://cn.string-db.org, accessed on 26 April 2023) was used to reveal a high correlation between p53 and TFRC, SLC39A14, and the ferroptosis-related signaling molecules (Figure 7A). The co-IP result showed that p53 interacted with SLC39A14 but not with TFRC (Figure 7B). The simulation and prediction molecular structure predicted the possibility of an interaction between p53 and SLC39A14, with a binding energy of −320.84 kcal/mol. The amino acid residues at the protein–protein interaction interface could form five hydrogen bonds to stabilize the p53–SLC39A14 complex (Figure 7C).

Treatment with proteasome inhibitor MG132 led to an accumulation of the SLC39A14 protein, suggesting the existence of a ubiquitination degradation pathway for the SLC39A14 protein (Figure 7D). Western blot analysis showed that the ubiquitination level of SLC39A14 was enhanced after p53 was overexpressed. The ubiquitinated SLC39A14 decreased after *Tfrc* knockdown. The ubiquitinated SLC39A14 was likewise increased in the *OE-Tfrc* group, confirming the direct relationship between p53 and SLC39A14 expression (Figure 7E). Immunofluorescence showed p53 and SLC39A14 co-localization in the cytoplasm around the nucleus (Figure A5), indicating that p53 can bind to SLC39A14 in the cytoplasm and promote its ubiquitination degradation.

We then proceeded to detect the content and location of the p53 protein. Western blot analysis showed that mitochondrial p53 increased but cytoplasmic p53 decreased in the *siTfrc* group (Figure 7F). Overlay fluorescence images demonstrated that the p53 protein was predominantly localized in the mitochondria of the cells in the *siTfrc* group, in contrast with the *siNC* group (Figure 7G). Moreover, the p53 level in the mitochondria was reduced while the p53 level in the cytoplasm was enhanced in the *OE-Tfrc* group (Figure 7H). Fluorescence images demonstrated that the p53 proteins were distributed primarily in the cytoplasm of the *OE-Tfrc* group (Figure 7I). The results indicated that the ablation of TFRC expression led to an increase in mitochondrial p53 translocation and a modulation of cytoplasmic p53, ultimately influencing the content of SLC39A14.

Mitochondrial p53 translocation was usually related to increased mitochondrial permeability. Using MitoFerroGreen, a fluorescent probe specific for Fe^2+^ in mitochondria, it was found that as the mitochondrial iron ions increased, more p53 entered the mitochondria in the *siTfrc* group (Figure A6).

To investigate whether mitochondrial stress was associated with ferroptosis, a calcein AM fluorescence probe was used to assess mitochondrial permeability. The results indicated that cells in the *siTfrc* group exhibited weaker fluorescence intensity compared to the control group, suggesting a higher number of open mitochondrial transition pores in the *siTfrc* group (Figure 7J). On the contrary, there were fewer mitochondrial transition pores open in the *OE-Tfrc* group than the control group (Figure 7K). MitoSOX Red staining revealed increased mitochondrial ROS production in the *siTfrc* group compared to the *siNC* group (Figure 7L). Overexpression of *Tfrc* effectively reduced mitochondrial ROS generation (Figure 7M).

This demonstrated that ferroptosis influenced mitochondrial permeability and facilitated mitochondrial p53 translocation following *Tfrc* knockdown. Mitochondrial p53 translocation is involved by the ubiquitin-mediated degradation of SLC39A14 in the chondrocytes, thereby affecting cartilage development and condylar growth.

### 2.9. Ferroptosis Inhibitor Fer1 Restores Chondrogenic Differentiation of ATDC5 Cells After Tfrc Knockdown

To further confirm the occurrence of ferroptosis and its role in decreased chondrogenic differentiation, Fer1, Ac-Met-OH and DFP were employed as rescue agents in vitro. The effects of inhibitors on cell proliferation and cytotoxicity were detected by the CCK-8 assay (Figure A7).

To determine whether ferroptosis contributes to *siTfrc*-induced chondrogenic dysfunction, we introduced Fer1, a potent and selective ferroptosis inhibitor, into the *siTfrc*-induced cell medium. Alcian blue staining of the micromass culture revealed that chondrogenic function was restored in the cells treated with Fer1 (Figure 8A). qPCR and Western blot analysis showed that the expression of SOX9 and ACAN was enhanced in the Fer1 group compared to the *siTfrc* group. The expression of the ferroptosis-related GPX4, SLC7A11 and FTH1 was increased while that of PTGS2, NOX1 and NCOA4 was decreased in the Fer1 group (Figure 8B,C). Flow cytometry analysis showed reduced ROS generation in the Fer1 group, with 43.98% DCFH-DA^+^ cells compared to 51.42% in the *siTfrc* group. The oxidation–reduction ratio was also more balanced in the Fer1 group (54.62% vs. 43.08%) compared to the *siTfrc* group (63.37% vs. 35.04%) (Figure 8D).

### 2.10. ROS Scavenger Ac-Met-OH and Iron Chelating Agent DFP Restore Chondrogenic Differentiation of ATDC5 Cells After Tfrc Knockdown

To alleviate the ROS generated by the Fenton reaction, cells in the *siTfrc* group were treated with Ac-Met-OH, a kind of ROS scavenger. Alcian blue staining of the micromass cultivation showed more active chondrogenesis in the cells treated with Ac-Met-OH (Figure 9A). qPCR analysis indicated the restored and increased expression of *Sox9*, *Acan* and *Gpx4*, while highly expressed genes *ptgs2*, *Ncoa4* and *Ascl4* were inhibited after Ac-Met-OH treatment (Figure 9B). Western blot analysis confirmed the suppression of PTGS2 and NCOA4 levels and the elevation of SOX9, ACAN, GPX4 and SLC7A11 levels in the Ac-Met-OH-treated cells (Figure 9C). Flow cytometry revealed 47.13% DCFH-DA^+^ cells in the Ac-Met-OH group, and the oxidation–reduction ratio, measured by the C11-BODITY fluorescence probe, was 61.01% vs. 36.92% in the Ac-Met-OH group (Figure 9D).

To restore intracellular iron homeostasis, we used DFP, a chelating agent that rapidly enters cells and clears iron ions from the cell. Alcian blue staining of the micromass cultivation showed enhanced chondrogenic differentiation in the DFP group (Figure 9A). qPCR analysis indicated that the expression of *Sox9*, *Acan* and *Gpx4* was restored and that the highly expressed genes *Ncoa4* and *Ascl4* were inhibited after treatment with DFP (Figure 9B). The levels of PTGS2, NOX1 and NCOA4 were suppressed and the levels of SOX9, ACAN, GPX4 and SLC7A11 in the DFP group were elevated, as identified by the Western blot analysis (Figure 9C). Flow cytometry showed 43.92% DCFH-DA^+^ cells in the DFP group, which was lower than in the *siTfrc* group. The oxidation–reduction ratio was also more balanced in the DFP group (55.96% vs. 42.11%) (Figure 9D). The above data show that Fer1, Ac-Met-OH and DFP could relieve oxidative stress, the cellular peroxidation status, as well as highlight the effect of DFP on promoting chondrogenic differentiation and relieving ferroptosis.

### 2.11. Iron Chelator DFP Rescues Mandibular (Condyle) Hypoplasia of Tfrc-cKO Mice

To ascertain the occurrence of ferroptosis in the condylar cartilage of mice and to perform an in vivo rescue assay, DFP was administered to the *Tfrc*-cKO mice through temporomandibular joint injections. Examination of the condyle in the *Tfrc*-cKO mice revealed degraded and loose bone trabeculae. Notably, following DFP treatment, the sub-cartilage bone trabeculae of the condyle exhibited thickening (Figure 10A). The length of the horizontal, vertical, and oblique axes of the condyle obtained a certain degree of recovery. The C-F length of the condyle of the control mice, cKO mice and *Tfrc*-cKO mice treated with DFP were 3.03 ± 0.11 mm, 2.64 ± 0.21 mm and 2.99 ± 0.23 mm, respectively. The mandibular lengths were 13.33 ± 1.06 mm, 11.19 ± 1.01 mm and 13.43 ± 0.44 mm, respectively. The other detailed data are shown in Figure 10B. IHC analysis showed that the levels of SOX9, ACAN, and anti-ferroptosis marker GPX4 decreased in the proliferative and hypertrophic layers of the *Tfrc*-cKO mice. However, these expressions could be significantly restored following the temporomandibular joint injection of DFP (Figure 10C). The semi-quantitative calculation results of IF and IHC are shown in Figure 10D. After the DFP treatment, SOX9, ACAN and GPX4 expression were increased at the mRNA and protein levels, along with decreased PTGS2 and NOX1 expression at the protein level in the condylar chondrocytes (Figure 10E,F). The results indicate that DFP rescues mandibular (condyle) hypoplasia of the *Tfrc*-cKO mice by eliminating iron overload-mediated ferroptosis.

## 3. Discussion

In this study, we intended to explore the role of TFRC on condylar cartilage development and the underlying mechanism of the TFRC–SLC39A14 switch in cartilage hypoplasia. Here, our study reveals that sTFRC reaches its peak during the pubertal stage of cartilage development. TFRC, which binds transferrin and facilitates its transfer, could promote the development of condyle cartilage. TFRC ablation would trigger a compensatory upregulation of SLC39A14, facilitating the transfer of NTBI into the cytoplasm. This switch in expression leads to ferroptosis in the chondrocytes, a process in which mitochondrial p53 translocation could mediate SLC39A14 ubiquitination degradation.

Initially, we investigated the molecular expression pattern of TFRC throughout cartilage development. Condyle growth during puberty is pivotal for orthodontic treatment timing and planning, particularly for individuals with SMH [9]. A previous study indicated that functional treatment via a removable appliance had a remarkable skeletal impact on correcting mandibular retrognathia at CS3-4 [28]. Because acquiring condylar tissue from adolescent patients is challenging, we obtained serum samples to reflect the local conditions based on systemic iron metabolism. Our study shows that sTFRC, the truncated form of TFRC on the surface of a cell, peaks at CS3 and that the SMH patient at CS3 exhibits a lower sTFRC level. The level of sTFRC positively correlates with iron demand [29,30]. Our animal experiments prove that TFRC fulfills a role in the postnatal condylar cartilage. Conversely, low expressed TFRC and TF with compensatory upregulated SLC39A14 and NTBI impedes condylar cartilage development. These findings also corroborate our group’s preliminary results that TF could be a candidate biomarker for detecting the peak period of mandibular growth in adolescent patients [15,27].

We have now identified ferroptosis in condylar cartilage development for the first time, both in the *Tfrc*-cKO mice and the *siTfrc*-induced chondrogenic dysfunction of the ATDC5 cells. Ferroptosis induced by the dominant expression switch of TFRC–SLC39A14 has been documented in both the liver and skeletal muscle tissues [31,32,33]. These cellular phenomena include NTBI overload, lipid peroxidation and reduced ROS clearance rate due to the destruction of the endogenous antioxidant system, resulting in reduced glutathione and GPX4 expression or increased positive ferroptosis-related biomarker expression including PTGS2 and NOX1. TFRC has been considered an important indicator of ferroptosis, bringing out temporomandibular arthritis by increased iron intake and an imbalance in the redox system [34]. However, our mice model yields different conclusions, indicating that the effects of the dual markers TFRC and SLC39A14 on iron homeostasis really matter together instead of TFRC alone.

In this study, we found that p53 is identified as a key factor for the TFRC–SLC39A14 expression switch. In response to the iron level and mitochondrial stress, p53 undergoes mitochondrial translocation and exerts its effect on post-translational modification [35]. Increased ROS generation leads to increased mitochondrial membrane permeability, allowing more p53 to enter into the mitochondria as previously reported [36]. Changes in the cytoplasmic p53 contribute to alterations in the ubiquitination degradation of the SLC39A14 protein, thereby regulating its content. However, we have not yet elucidated the underlying mechanism by which changes in the SLC39A14 expression lead to alterations in the TFRC expression. It may be because a decrease in NTBI transport via SLC39A14 triggers a compensatory increase in TFRC that could facilitate TBI transport.

Our findings suggest the potential of DFP in treating condyle growth downregulation or augmenting the therapeutic effects of functional appliances. To counteract iron accumulation, ROS production and lipid peroxidation, we used the ferroptosis inhibitor Fer1, ROS scavenger Ac-Met-OH and iron chelating agent DFP to rescue insufficient chondrogenic differentiation in vitro. The results aligned with our expectations. Ferroptosis inhibitors such as Fer1 and deferoxamine could alleviate osteoarthritis [34,37]. In our study, DFP, approved by the Food and Drug Administration (FDA) for treating iron overload-related diseases [38], is used to treat deficient mandibular condyle and achieve certain therapeutic effects. However, due to the non-specific nature of DFP, the precise effects of DFP on other organs remain undetermined, and its potential side effects need to be further investigated.

## 4. Materials and Methods

### 4.1. Study Approval

All patients’ samples were collected under the approval from the Biomedical Ethics Committee of Peking University School and Hospital of Stomatology (approval number: PKUSSIRB-202055072). Animal experiments were conducted in accordance with the Guide for the Care and Use of Laboratory Animals with approval of Biomedical Ethics Committee of Peking University (approval number: LA2021538).

### 4.2. Participants and Clinical Evaluations

The SMH patients and normal control subjects were recruited from a pool of individuals seeking orthodontic treatment in the Peking University School and Hospital of Stomatology from July 2020 to January 2021. Both the patients and their parents provided informed consent. Lateral cephalograms were captured during their initial visit based on the treatment requirements. The cervical vertebral maturation stage (CVM stage, CS) is an indicator to identify the peak of mandible growth and the optimal timing of growth modification. According to a previous study [39], CVM stage is determined through the analysis of the second to fourth cervical vertebrae in a single cephalogram, and the CVM stage at 3–4 (CS 3–4) is considered as the indicative biomarker of the pubertal growth peak. The following markers are defined in the lateral cephalograms: N, Nasion; A, Subspinale; B, Supramental. ANB angle represents the imbalance between maxilla and mandible. SMH patients were diagnosed based on the following criteria: 1. ANB> 5°; 2. Overjet > 3 mm; 3. Distal molar relationship; and 4. No systemic disease. Normal control subjects were defined according to the criteria as follows: 1. 0 < ANB < 5°; 2. 0 < Overjet < 3 mm; 3. Neutral molar relationship; and 4. No systemic disease.

### 4.3. Serum Analyses and Iron Metabolism Determination

During the patients’ initial visit between 8:00 a.m. and 10:00 a.m., approximately 4 mL of venous blood was collected from each subject. The methods for serum separation and calculation of minimum sample size were described in a previous study [27]. The detection of iron-related indicators in serum can serve as an indicator of the iron metabolism status in tissues [30]. Serum analysis and iron metabolism determination were conducted by the Peking University Third Hospital.

### 4.4. Generation of Col2-Cre^ERT^; Tfrc^fl/fl^ Mice and Treatments

Col2-Cre^ERT^ mice and *Tfrc*^fl/fl^ mice were crossed to generate Col2-Cre^ERT^; *Tfrc*^fl/fl^ genotype mice. The genotype Col2-Cre^ERT^; *Tfrc*^fl/fl^ was designated as homozygous (*Tfrc*-cKO) and *Tfrc*^fl/fl^ was designated as control (Ctrl). Chondrocyte-specific *Tfrc* knockout mice, without distinguishing between males and females, were generated by intraperitoneal injection of tamoxifen (TMX) dissolved in corn oil at a dose of 15 mg/mL for seven consecutive days, starting one week after birth. All experimental mice were placed on a 12 h/12 h light/dark cycle with free access to food and water. For animal welfare, all mice were sacrificed by pentobarbital sodium overdose.

### 4.5. Immunofluorescence (IF)

The condyles were removed from the normal mice at 0, 4, 8 and 12 weeks after birth or from the *Tfrc*-cKO and Ctrl mice at 4 weeks. The samples were fixed in 4% paraformaldehyde, demineralized in 10% ethylenediaminetetraacetic acid, dehydrated in graded alcohol and xylene, embedded in paraffin, and sectioned to a thickness of 5 μm. The sections were blocked with 5% bovine serum albumin and incubated with primary antibodies against TFRC (1:100, ab214039, Abcam, Shanghai, China) at 4 °C overnight. After washing with phosphate-buffered saline (PBS), the sections were incubated with the secondary fluorescence antibody for 1 h. The nuclei were stained with fluorescent mounting medium (ZLI-9556, ZSGB-BIO, Beijing, China) on a glass slide. The image was revealed by a fluorescence microscope. The semi-quantitative analysis of IF was based on the previous literature [40].

### 4.6. Immunohistochemistry (IHC) and Haematoxylin and Eosin (H&E) Staining

The condyles were removed from normal mice at 0, 4, 8 and 12 weeks after birth or from the *Tfrc*-cKO mice and the Ctrl mice at 4 weeks. The samples were subjected to IHC and processed similarly to those for IF, including fixation, demineralization, dehydration, embedding and sectioning. The sections were blocked with 5% bovine serum albumin and incubated with primary antibodies against transferrin (TF, 1:100, 17435-1-AP, Proteintech, Chicago, IL, USA), SLC39A14 (1:200, DF14224, Affinity Biosciences, Liyang, Jiangsu, China), aggrecan (ACAN, 1:200, DF7561, Affinity Biosciences, Liyang, Jiangsu, China), SOX9 (1:200, AF6330, Affinity Biosciences, Liyang, Jiangsu, China) and GPX4 (1:200, DF6701, Affinity Biosciences, Liyang, Jiangsu, China) at 4 °C overnight. After washing with PBS, the sections for IHC were incubated with HRP-conjugated secondary antibodies, and the staining was revealed with diaminobenzidine. Haematoxylin and eosin (H&E) staining was performed according to the instructions of the H&E staining kit (G1120, Solarbio Science and Technology Pty, Ltd., Beijing, China). The semi-quantitative analysis of IHC was based on the previous literature [40].

### 4.7. Saffron O and Fast Green Staining

Saffron O and Fast Green staining was used to distinguish the cartilage (stained red by Saffron O, a basic dye) and bone (stained green or blue by Fast Green, an acid dye) tissues. The mandibular condyles were removed from the normal mice at 0, 4, 8 and 12 weeks after birth or from the *Tfrc*-cKO mice at 4 weeks, fixed in 4% paraformaldehyde and demineralized in 10% EDTA solution. The samples were processed similarly to those for IF and IHC, including dehydration, embedding and sectioning. Saffron-O and fast green staining was performed according to the instructions provided with Modified Saffron O and Fast Green Staining Kit (G1371, Solarbio Science and Technology Pty, Ltd., Beijing, China).

### 4.8. Cell Culture and Micromass Culture

The mouse chondrogenic progenitor cell line ATDC5 was used to investigate the chondrogenic differentiation process. The cells were purchased from the American Type Culture Collection (ATCC, Manassas, VA, USA) and cultured under aseptic conditions using high-glucose Dulbecco’s modified Eagle medium (DMEM, Hyclone, Logan, GA, USA) containing 10% fetal bovine serum (FBS, Biological Industries, Kibbutz Beit Haemek, Israel) and 50 U/mL penicillin and 50 µg/mL streptomycin (Hyclone, Logan, GA, USA) in a humidified atmosphere of 5% CO_2_ at 37 °C. The cells were divided into samples, with each containing approximately 5 × 10^3^ cells in a 40 μL volume, and placed into 12-well culture plates for micromass culture. The cells were incubated at 37 with 5% CO_2_ for 2 h to create a micromass culture. Next, the culture medium containing 1× insulin-transferrin-selenium (ITS, Yishan Biotech Co., Shanghai, China) was added without cell spreading [41].

### 4.9. siRNA and Overexpression Plasmid Transfection

To knock down the *Tfrc* expression, small interfering RNA (siRNA) specially targeting *Tfrc* was designed (sense, GGAUAUGGGUCUAAGUCUATT, antisense, UAGACUUAGACCCAUAUCCTT) and transfected into the ATDC5 cells using the JetPRIME transfection reagent (Ployplustransfection, Illkirch, France)*. siNC* was used as a negative control (Shanghai Lingke Tech, Shanghai, China). The cells were incubated for 48 h after transfection. To knock down the *Slc39a14* expression, another siRNA specially targeting *Slc39a14* was designed (sense, GCUUCAACCCUCAGGACAATT, antisense, UUGUCCUGAGGGUUGAAGCTT).

To overexpress *Tfrc* or *p53*, the Flag-tagged *Tfrc* overexpression plasmid and the Flag-tagged *p53* overexpression plasmid (Flag tag at the C-terminal) were constructed by Tsingke Biotechnology Co., Ltd., Beijing, China. The plasmids were extracted by the EndoFree Plasmid Midi Kit (Jiangsu Cowin Biotech Co., Taizhou, Jiangsu, China) and transfected into the ATDC5 cells using JetPRIME transfection reagent. The cells were cultured for 24 h after transfection for the subsequent assays. Cells transfected with an empty vector (*OE-EV*) were used as negative control.

### 4.10. Chondrogenic Differentiation Induction

At 48 h post-transfection, the medium was replaced with a chondrogenic-induced medium supplemented with ITS (Yishan Biotech Co., Shanghai, China) and a reduced level of FBS to induce serum deprivation. The medium was changed every day for the 5–7 days of chondrogenic induction.

### 4.11. Alcian Blue Staining

After 7 days of chondrogenic differentiation, the cell cultures were washed three times with PBS and fixed with 4% paraformaldehyde for 10 min. Chondrogenic differentiation was assessed using Alcian blue staining using the Alcian Blue 8GX Kit (Solarbio, Beijing, China).

### 4.12. Cell Proliferation and Cytotoxicity Assay

Cell viability was assessed using the CCK-8 Assay Kit (Dojindo, Kyushu, Japan), following the manufacturer’s protocol. In this study, the ferroptosis inhibitor ferrostatin-1 (Fer1, 1 μM, dissolved in DMSO, Selleck Chemicals, Shanghai, China), the ROS scavenger N-Acetyl-L-methionine (Ac-Met-OH, 600 μM, dissolved in DMSO, Selleck Chemicals, Shanghai, China) and the iron chelating agent deferiprone (DFP, 300 μM, dissolved in water) were employed in the cell proliferation and cytotoxicity assays. The concentrations of these inhibitors were determined based on the previous studies to ensure that they did not significantly inhibit cell growth [20,38].

### 4.13. Reduced Glutathione (GSH) Assay and Iron Assay

The cellular reduced glutathione level was quantified by the Reduced Glutathione Assay Kit (Jiancheng, Nanjing, Jiangsu, China) and read at 405 nm according to the instructions. Additionally, cellular iron content was quantitatively determined by employing the Iron Assay Kit (Jiancheng, Nanjing, Jiangsu, China), with readings taken at 520 nm in accordance with the provided instructions. The difference in iron content displayed by the iron assay and TF expression displayed by the Western blot can be considered as the content of NTBI.

### 4.14. Reactive Oxygen Species Assay and Lipid Peroxidation Detection

Intracellular reactive oxygen species (ROS) levels were detected using the Reactive Oxygen Species Assay Kit (S0033S, Beyotime, Shanghai, China), following the manufacturer’s protocol. Lipid peroxidation was detected utilizing BODIPY™ 581/591 C11(D3861, Invitrogen, Carlsbad, CA, USA). After incubating the cells at 37 °C for 30 min, they were examined within 1 h using a flow cytometer (NovoCyte, Agilent, Santa Clara, CA, USA) according to the manufacturer’s instructions.

### 4.15. MitoFerroGreen Staining and Mitochondrial Staining

To assess the intracellular iron ion content, MitoFerroGreen (Dojindo, Kyushu, Japan) was utilized. Adherent cells, plated on coverslips, were incubated with 5 μmol/L MitoFerroGreen for 30 min. Subsequently, the nuclei were stained using a fluorescent mounting medium (ZLI-9556, ZSGB-BIO, Beijing, China) on a glass slide, and the cells were observed under TCS-SP8 DIVE (Leica, Wetzlar, Germany). For imaging mitochondria, MitoTracker Red CMXRos (Solarbio, Beijing, China) was employed according to the instructions provided. Adherent cells on coverslips were incubated with 100 nM MitoTracker red for 30 min allowing for the observation of mitochondrial morphology through TCS-SP8 DIVE.

Following the MitoFerroGreen or mitochondrial staining, the cells underwent immunofluorescence assays. They were washed three times with PBS and fixed with 4% paraformaldehyde for 10 min. The cells were permeabilized by 0.1% Triton X-100 for 10 min, blocked with BSA buffer for 1 h, and probed overnight with 1:100 diluted anti-p53 antibody (ab26, Abcam Shanghai Trading Co., Ltd., Shanghai, China) and anti-SLC39A14 antibody. The cells were treated with the corresponding secondary antibodies. The nuclei were stained with a fluorescent mounting medium on a glass slide. The cells were observed through a fluorescence microscope.

### 4.16. Mitochondrial Permeability Transition Pore (MPTP) and Mitochondrial Superoxide Detection

The MPTP was tested using the MPTP Assay Kit (C2009S, Beyotime, Shanghai, China) and mitochondrial superoxide was detected using the MitoSOX Red Mitochondrial Superoxide Indicator (MX4313-50UG, Maokang Biotechnology Co., Shanghai, China), both according to the manufacturer’s instructions.

### 4.17. Quantitative Real-Time PCR (qPCR)

After 3–5 days of in vitro chondrogenesis induction, the expression of the chondrogenic genes, *Sox9* and *Acan,* and the ferroptosis-related genes, *Tfrc*, *Slc39a14*, *Gpx4*, *Ptgs2*, *Ncoa4*, *Nox1*, *Ascl4*, *Fth1* and *Slc7a11,* was detected at the transcriptional level. Briefly, the total RNA of the cells and bone tissue was extracted using the TRIzol reagent according to the manufacturer’s instruction (Invitrogen, Carlsbad, CA, USA). Prior to RNA extraction, the bone tissue was crushed for 2 min using a high-throughput grinder, Tissuelyser II (QIAGEN, Germantown, MD, USA). Primers were designed using Primer3 4.1.0 and BLAST 2.16.0 to amplify specific regions within the coding sequences of the target genes and ribosomal protein S18 (*Rps18*, housekeeping control), ranging from 80 to 220 bp. The primer sequences and the expected sizes of the PCR products are shown in Appendix A Table A3. All PCR reactions were conducted in a 7500 Real-Time PCR machine using Taq Pro Universal SYBR Green qPCR Master Mix (Vazyme, Nanjing, Jiangsu, China). The PCR parameters for the target genes were as follows: 40 cycles for denaturation at 95 °C for 5 s, and then extension at 60 °C for 1 min. The expression levels for each gene of interest were normalized to their corresponding *Rps18* values. The comparative threshold cycle method was applied in the quantitative real-time PCR (qPCR) assay according to the 2^−∆∆Ct^ threshold cycle method. Each run was replicated at least three times.

### 4.18. Western Blot Analysis

Protein was extracted with a cell lysis buffer for Western blot and IP analysis (Solarbio, Beijing, China), supplemented with 1% PMSF (Solarbio, Beijing, China) and 1% protein phosphatase inhibitor (Huangxingbio, Beijing, China). Prior to protein extraction, the cartilage tissue was homogenized for 2 min by Tissuelyser II. Mitochondrial and cytoplasmic protein was separately extracted using the Cell Mitochondria Isolation Kit (Beyotime, Shanghai, China). Protein concentrations were determined by the BCA Protein Assay Kit (Solarbio, Beijing, China), with readings taken at 562 nm. Equal amounts of protein were fractionated by electrophoresis on a 10% SDS-PAGE gel and electro-transferred to the PVDF membrane. Then, the protein was probed with antibodies specific to TFRC, TF, SLC39A14, SOX9, ACAN, GPX4, PTGS2(AF7003, Affinity Biosciences, Liyang, Jiangsu, China), NCOA4 (DF4255, Affinity Biosciences, Liyang, Jiangsu, China), NOX1 (DF8684, Affinity, Liyang, Jiangsu, China) and FTH1 (DF6278, Affinity Biosciences, Liyang, Jiangsu, China). The level of protein expression was normalized relative to the control, RPS18 (DF3679, Affinity Biosciences, Liyang, Jiangsu, China), or GAPDH (AF7021, Affinity Biosciences, Liyang, Jiangsu, China).

### 4.19. Co-Immunoprecipitation (Co-IP)

Protein extraction was carried out using a cell lysis buffer intended for Western blotting and IP analysis, supplemented with 1% PMSF and 1% protein phosphatase inhibitor. Following thorough lysis and centrifugation, the supernatant was isolated and treated with RNase A (10 μg/mL) to eliminate mRNA interference. Then, a portion of the supernatant was aliquoted into a 1.5 mL centrifuge tube and mixed with 1/4 volume of 5× protein loading buffer. This mixture was denatured at 99 °C for 10 min and designated as the Input group. The remaining supernatant was evenly divided into two 1.5 mL centrifuge tubes and incubated with IgG and the corresponding antibodies overnight at 4 °C. The next step involved adding 30 μL of Protein A/G beads (TransGen Biotech, Beijing, China) and incubating the samples for 3–4 h. After incubation, the beads were washed with pre-cooled cell lysis buffer 3 times to eliminate non-specific binding. After centrifuging the samples and obtaining the supernatant, 2× protein loading buffer was added to the beads, and the mixture was heated at 99 °C for 10 min. Finally, Western blot analysis was conducted to ascertain the interaction between the two proteins.

### 4.20. Ubiquitination Detection

Prior to sample collection, proteasome inhibitor MG132 (S2619, Selleck, Huston, TX, USA) was introduced into the cell culture medium for 5 h, following transfection and chondrogenic differentiation. The subsequent steps mirrored those of the co-IP and Western blot procedures. The protein was then probed using an antibody specific to ubiquitin (Ub, 20326, Cell Signaling Technology, Danvers, MA, USA).

### 4.21. Temporomandibular Joint Injection of DFP Rescues Mice In Vivo

In the treatment groups, the temporomandibular joint injection of deferiprone (DFP) was administered every 12 h at a 10 mg/kg dose, dissolved in 0.9% normal saline [42]. This regimen commenced one week after TMX had been used to induce the *Tfrc* conditional knockout in the Col2-Cre^ERT^; *Tfrc*^fl/fl^ genotype mice and continued for 7 days. The mice were euthanized after 4 weeks using an overdose of pentobarbital sodium, adhering to the animal welfare guidelines.

### 4.22. Micro-CT Scanning and Analysis

Mandible samples from at least three mice per group, blinded to the researchers, were selected and counted by three experienced individuals. Each group had a minimum sample size of 6. The mandible samples were scanned using micro-CT (SkyScan1276, Bruker, Beijing, China), with the condition of 70 kV and 200 mA. Two-dimensional images were used to generate three-dimensional reconstructions. Consistent filtering and segmentation values were applied across all measurements to obtain three-dimensional images. Identical regions of the condyles were chosen as regions of interest (ROIs) for evaluation by micro-CT CT-Analyser software (Bruker, Beijing, China). The parameters were scored on the three-dimensional (3D) mandible reconstructions of the mice using the CT-Analyser 1.20.8.0 and CT-vox software 3.3 (Bruker, Beijing, China). The sagittal and coronal micro-CT scans were obtained from the Data-viewer software 1.7.0.1 (Bruker, Beijing, China).

### 4.23. Statistical Analysis

The number of repeats for each experiment is indicated in the figure legends. Data evaluation was conducted using GraphPad Prism, version 6.0 (GraphPad Software, San Diego, CA, USA). All samples were included in the analysis unless reliable results could not be obtained because of decomposition or contamination. The data are presented as mean ± SD. Statistical analysis was determined using one-way ANOVA followed by Tukey’s honest significant difference post hoc test for comparing more than 2 groups or two-tailed Student’s *t* test for comparing 2 groups. *p* < 0.05 was set as the statistical significance.

## 5. Conclusions

In this study, we aimed to delve into the role of TFRC in condylar cartilage development and the mechanism of the TFRC–SLC39A14 switch in cartilage hypoplasia. We have determined that TFRC expression peaks during pubertal mandible growth, and TFRC ablation leads to inadequate cartilage development by the upregulation of SLC39A14. The TFRC–SLC39A14 expression switch initiates the ferroptosis of chondrocyte, characterized by NTBI accumulation, lipid peroxidation and antioxidant mechanism disruption. Decreased ubiquitination degradation of SLC39A14 mediated by mitochondrial p53 translocation leads to the TFRC–SLC39A14 expression switch. In addition, the iron chelating agent DFP could rescue the cartilage phenotype by inhibiting ferroptosis. Our study not only unveils a novel mechanism but also offers new perspectives on the etiology, pathogenesis and treatment of skeletal mandibular hypoplasia.

## Figures and Tables

**Figure 1 ijms-26-02724-f001:**
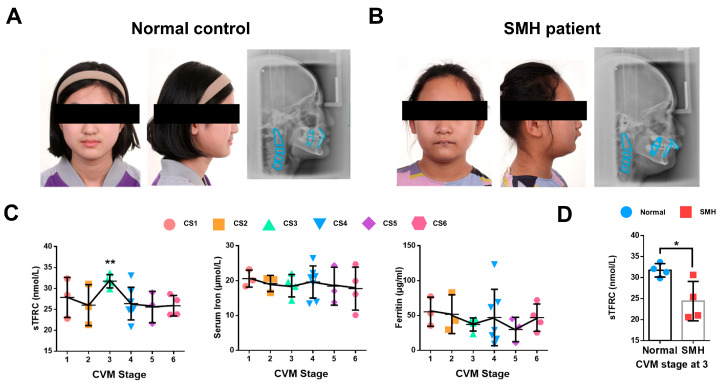
The expression of sTFRC decreases in SMH patients at CVM stage 3. (**A**) Facial image and lateral cephalogram of a female subject without SMH. (**B**) Facial image and lateral cephalogram of a female patient with SMH. (**C**) Serum TFRC (sTFRC), serum iron and ferritin levels at CS1–6. The data points of different colors were covering CS1-6 from left to right in turn. (**D**) sTFRC level of SMH and normal subjects at CS3 (*n* = 4, unpaired two-tailed Student’s *t* test). Blue circle represents data of normal subjects and red square represents data of SMH patients. Data are presented as mean ± SD. *: *p* < 0.05, **: *p* < 0.01.

**Figure 2 ijms-26-02724-f002:**
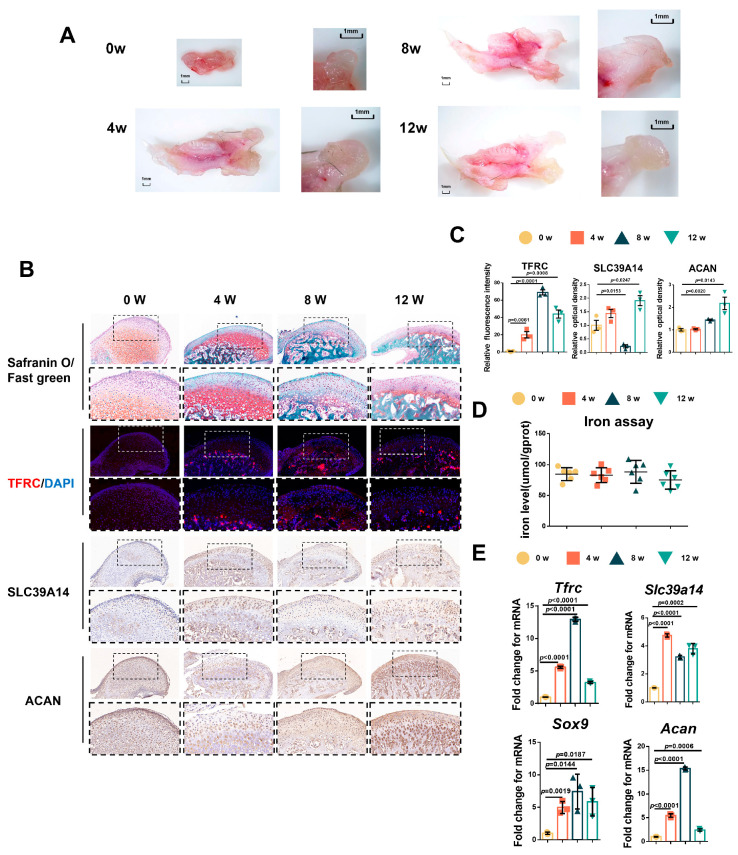
The relationship between condylar cartilage development and the expression of TFRC and SLC39A14. (**A**) The gross morphology of mandibular condyles from newborn mice to 12 w mice. (**B**) Representative safranin O/fast green staining images of the condyle from 0 w to 12 w mice. IF staining of TFRC expression in the condyle from 0 w to 12 w mice. IHC staining of SLC39A14 and ACAN expression in the condyle from 0 w to 12 w mice. The box encircled the representative area encompassing the proliferative and hypertrophic layers of the cartilage. The image with a dashed border below is an enlarged image of the image above. (**C**) The semi-quantitative calculation results of IF and IHC (*n* = 3, one-way ANOVA). (**D**) Total iron detected in condylar cartilage from 0 w to 12 w mice (*n* = 6, one-way ANOVA). (**E**) qPCR analysis of *Tfrc, Slc39a14, Sox9* and *Acan* expression in condylar cartilage from 0 w to 12 w mice (*n* = 3, one-way ANOVA). Yellow circle represents data of 0 w group. Orange square represents data of 4 w group. Dark green triangle represents data of 8 w group. Light green triangle represents data of 12 w group. Data are presented as mean ± SD.

**Figure 3 ijms-26-02724-f003:**
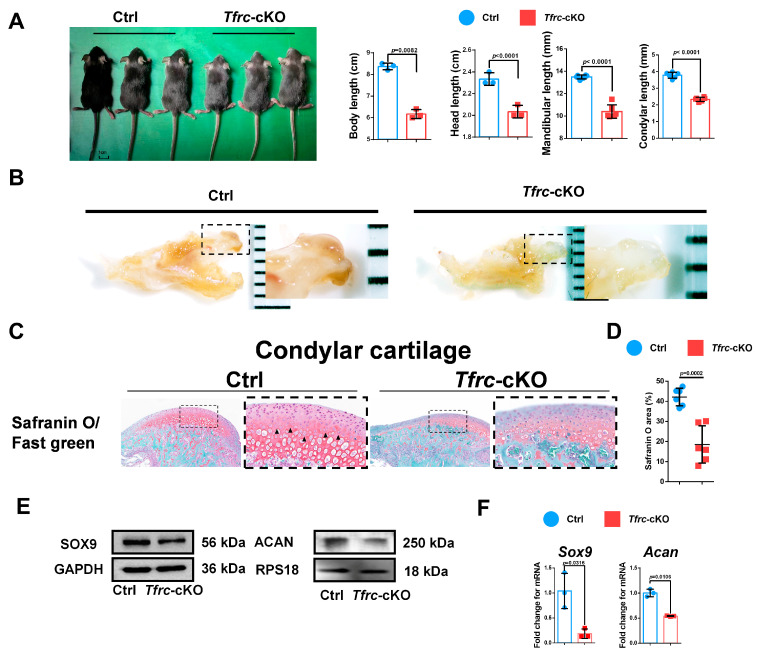
Generation of chondrogenic conditional knockout mice with weaker condylar chondrogenic differentiation ability. (**A**) Gross morphology of mice in Ctrl and *Tfrc*-cKO group. The length measurement of the body excluding the tail, the head, the mandible and the condyle in Ctrl and *Tfrc*-cKO mice group. Mean ± SD (mice *n* = 6, mandible *n* = 6, unpaired two-tailed Student’s *t* test). (**B**) The gross morphology of chondyle in the Ctrl and *Tfrc*-cKO mice. (**C**) Representative safranin O/fast green staining images of the condyle of Ctrl and *Tfrc*-cKO mice. The image with a dashed border on the right is an enlarged image of the left image. (**D**) The semi-quantitative calculation results of safranin O^+^ area (*n* = 3, unpaired two-tailed Student’s *t* test). (**E**) Western blot analysis of SOX9 and ACAN expression in the condylar cartilage of Ctrl and *Tfrc*-cKO mice. (**F**) qRCR analysis of *Sox9* and *Acan* expression in the condylar cartilage of Ctrl and *Tfrc*-cKO mice (*n* = 3, unpaired two-tailed Student’s *t* test). Blue circle represents data of Ctrl group and red square represents data of *Tfrc*-cKO. Data are presented as mean ± SD.

**Figure 4 ijms-26-02724-f004:**
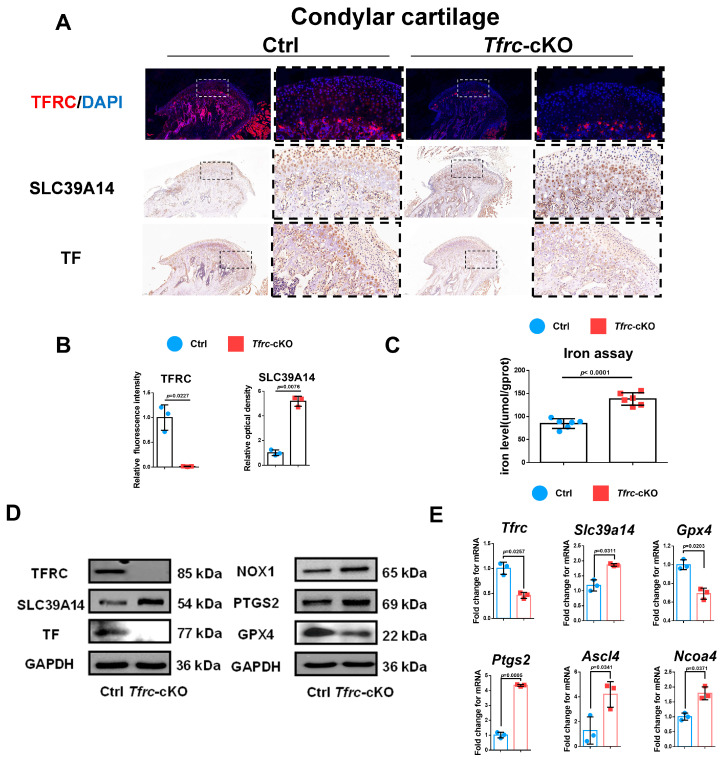
Negative correlation expression between TFRC and SLC39A14 in condylar cartilage of *Tfrc*-cKO mice. (**A**) IF staining of TFRC expression in the condyle of Ctrl and *Tfrc*-cKO mice. IHC staining of SLC39A14 and TF expression in the condyle of Ctrl and *Tfrc*-cKO mice. The image with a dashed border on the right is an enlarged image of the left image. (**B**) The semi-quantitative calculation results of IF and IHC (*n* = 3, unpaired two-tailed Student’s *t* test). (**C**) Total iron detected in condylar cartilage of Ctrl and *Tfrc*-cKO mice (*n* = 6, unpaired two-tailed Student’s *t* test). (**D**) Western blot analysis of TFRC, SLC39A14, TF, GPX4, PTGS2 and NOX1 expression in the condylar cartilage of Ctrl and *Tfrc*-cKO mice. (**E**) qRCR analysis of *Tfrc*, *Slc39a14*, *Gpx4*, *Ptgs2*, *Ascl4* and *Ncoa4* expression in the condylar cartilage of Ctrl and *Tfrc*-cKO mice (*n* = 3, unpaired two-tailed Student’s *t* test). Blue circle represents data of Ctrl group and red square represents data of *Tfrc*-cKO group. Data are presented as mean ± SD.

**Figure 5 ijms-26-02724-f005:**
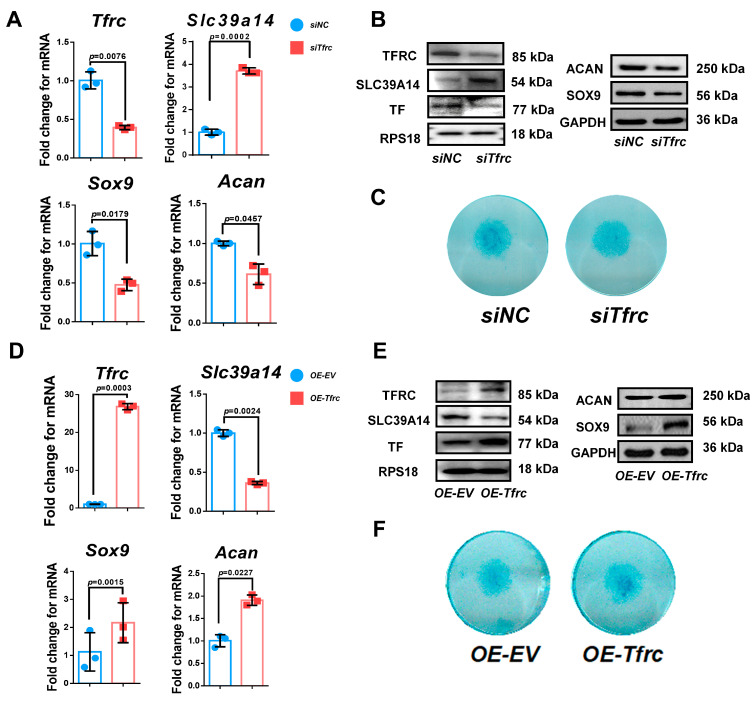
TFRC promotes chondrogenic differentiation of ATDC5 cells through negatively regulating SLC39A14 expression. (**A**) qPCR analysis of *Tfrc*, *Slc39a14*, *Sox9* and *Acan* expression in *siNC* and *siTfrc* groups (*n* = 3, unpaired two-tailed Student’s *t* test). Blue circle represents data of *siNC* group and red square represents data of *siTfrc* group. (**B**) Western blot analysis of TFRC, TF, SLC39A14, SOX9 and ACAN expression in *siNC* and *siTfrc* groups. (**C**) ATDC5 cells of micromass cultivation stained with Alcian blue in *siNC* and *siTfrc* groups. (**D**) qPCR analysis of *Tfrc*, *Slc39a14*, *Sox9* and *Acan* expression in *OE-EV* and *OE-Tfrc* groups (*n* = 3, unpaired two-tailed Student’s *t* test). Blue circle represents data of *OE-EV* group and red square represents data of *OE-Tfrc* group. (**E**) Western blot analysis of TFRC, TF, SLC39A14, SOX9 and ACAN expression in *OE-EV* and *OE-Tfrc* groups. (**F**) ATDC5 cells of micromass cultivation stained with Alcian blue in *OE-EV* and *OE-Tfrc* groups. Data are presented as mean ± SD.

**Figure 6 ijms-26-02724-f006:**
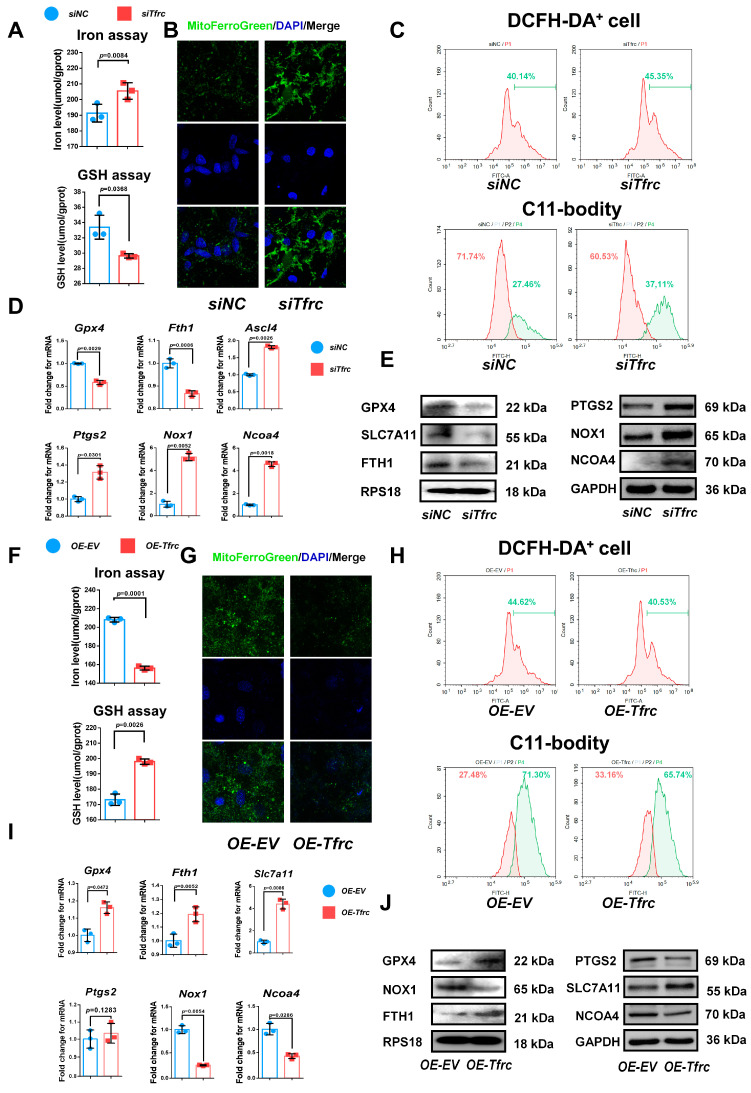
TFRC promotes chondrogenic differentiation of ATDC5 cells by regulating ferroptosis. (**A**) Total iron and glutathione detection in *siNC* and *siTfrc* groups (*n* = 3, unpaired two-tailed Student’s *t* test). Blue circle represents data of *siNC* group and red square represents data of *siTfrc* group. (**B**) Confocal microscopy images of MitoFerroGreen staining demonstrating iron accumulation in mitochondria in *siNC* and *siTfrc* groups. (**C**) The proportion of DCFH-DA^+^ cells and ratio of oxidation and reduction cells detected by flow cytometry in *siNC* and *siTfrc* groups. (**D**) qPCR analysis of *Ptgs2*, *Ascl4*, *Nox1*, *Ncoa4*, *Gpx4* and *Fth1* expression in *siNC* and *siTfrc* group (*n* = 3, unpaired two-tailed Student’s *t* test). Blue circle represents data of *siNC* group and red square represents data of *siTfrc* group. (**E**) Western blot analysis of GPX4, SLC7A11, FTH1, PTGS2, NOX1 and NCOA4 expression in *siNC* and *siTfrc* groups. (**F**) Total iron and glutathione detection in *OE-EV* and *OE-Tfrc* groups (*n* = 3, unpaired two-tailed Student’s *t* test). Blue circle represents data of *OE-EV* group and red square represents data of *OE-Tfrc* group. (**G**) Confocal microscopy images of MitoFerroGreen staining demonstrating iron accumulation in mitochondria of *OE-EV* and *OE-Tfrc* groups. (**H**) The proportion of DCFH-DA^+^ cells and ratio of oxidation and reduction cells detected by flow cytometry in *OE-EV* and *OE-Tfrc* groups. (**I**) qPCR analysis of *Gpx4*, *Slc7a11*, *Fth1, Ptgs2*, *Nox1* and *Ncoa4* in *OE-EV* and *OE-Tfrc* groups (*n* = 3, unpaired two-tailed Student’s *t* test). Blue circle represents data of *OE-EV* group and red square represents data of *OE-Tfrc* group. (**J**) Western blot analysis of GPX4, SLC7A11, FTH1, PTGS2, NOX1 and NCOA4 expression in *OE-EV* and *OE-Tfrc* groups. Data are presented as mean ± SD.

**Figure 7 ijms-26-02724-f007:**
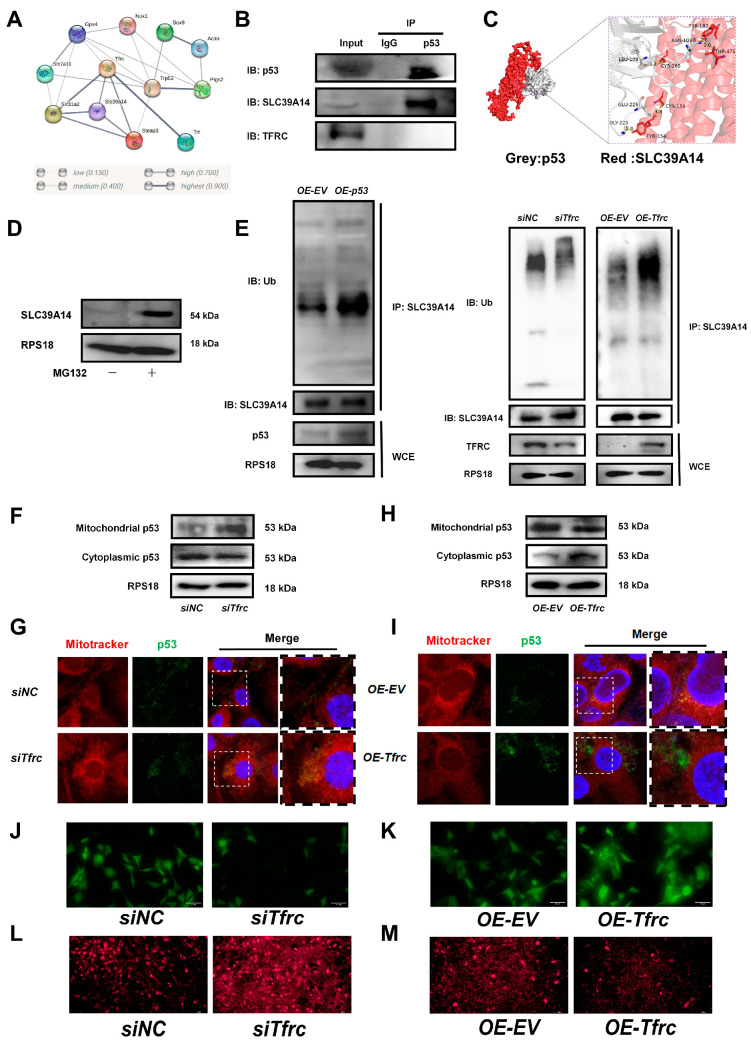
The FRC–SLC39A14 expression switch regulates ferroptosis through mitochondrial p53 translocation in ATDC5 cells (**A**) Using the String website (cn.string-db.org) to predict the correlation between protein molecules, it was found that p53 were highly correlated with TFRC, SLC39A14, and ferroptosis-related signaling molecules. (**B**) co-IP analysis of protein interaction between p53, SLC39A14 and TFRC. (**C**) Using Hdock (hdock.phys.hust.edu.cn) to dock ZIP14 and p53 interaction. The result of molecular docking showed that the binding energy of p53 and SLC39A14 was −320.84 kcal/mol. The residues around the protein–protein interaction interface could form hydrogen bonds. The interaction mode of docking results was analyzed by Pymol 2.3.0. The amino acid residue GLY-223, GLU-225, LEU-108 and ASN-103 of p53 formed hydrogen bonds with TYR-154, CYS-158, CYS-165, THR-471 and TYR-187 of SLC39A14, respectively. The length of hydrogen bonds was 2.0Å,3.8Å, 3.4Å, 2.8Å and 3.0Å. These hydrogen bonds could help stabilize protein–protein complexes. (**D**) Western blot analysis of SLC39A14 expression after treatment with proteasome inhibitor MG132. (**E**) Western blot analysis of ubiquitination level of SLC39A14 after *p53* overexpression, *Tfrc* overexpression and *Tfrc* knockdown. (**F**) Western blot analysis of mitochondrial p53 and cytoplasmic p53 level in *siNC* and *siTfrc* groups. (**G**) Fluorescence images of cells and mitochondria to detect p53 translocation in *siNC* and *siTfrc* groups. (**H**) Western blot analysis of mitochondrial p53 and cytoplasmic p53 level in *OE-EV* and *OE-Tfrc* groups. (**I**) Fluorescence images of cells and mitochondria to detect p53 translocation in *OE-EV* and *OE-Tfrc* groups. (**J**) Calcein AM fluorescence probe was used to detect mitochondrial permeability in *siNC* and *siTfrc* groups. (**K**) Calcein AM fluorescence probe was used to detect mitochondrial permeability in *OE-EV* and *OE-Tfrc* groups. (**L**) The images of fluorescence microscope detecting ROS (MitoSOX red) generated by mitochondria in *siNC* and *siTfrc* groups. (**M**) The images of fluorescence microscope detecting ROS (MitoSOX red) generated by mitochondria in *OE-EV* and *OE-Tfrc* groups.

**Figure 8 ijms-26-02724-f008:**
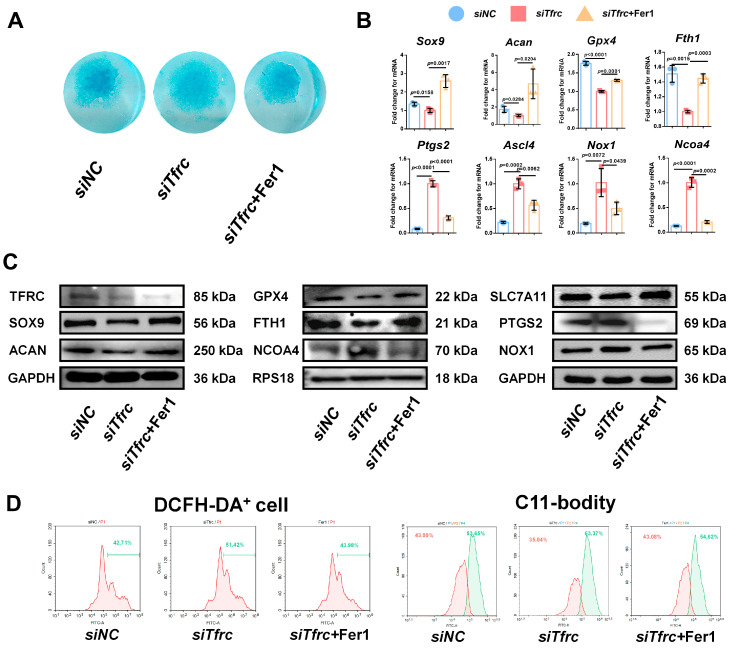
Ferroptosis inhibitor Fer1 restores chondrogenic differentiation of ATDC5 cells after *Tfrc* knockdown. (**A**) Alcian blue staining of micromass culture in *siNC* group, *siTfrc* group and cells treated with Fer1. (**B**) qPCR results of *Sox9*, *Acan*, *Gpx4*, *Fth1*, *Ptgs2*, *Ascl4*, *Nox1* and *Ncoa4* after treatment with Fer1 (*n* = 3, one-way ANOVA). Blue circle represents data of *siNC* group, red square represents data of *siTfrc* group and orange triangle represents data of *siTfrc* + Fer1 group. (**C**) Western blot analysis of TFRC, SOX9, ACAN, GPX4, SLC7A11, FTH1, PTGS2, NOX1 and NCOA4 in *siNC*, *siTfrc* and *siTfrc* + Fer1 groups. (**D**) The proportion of DCFH-DA^+^ cells and ratio of oxidation and reduction cells detected by flow cytometry in *siTfrc* + Fer1 group.

**Figure 9 ijms-26-02724-f009:**
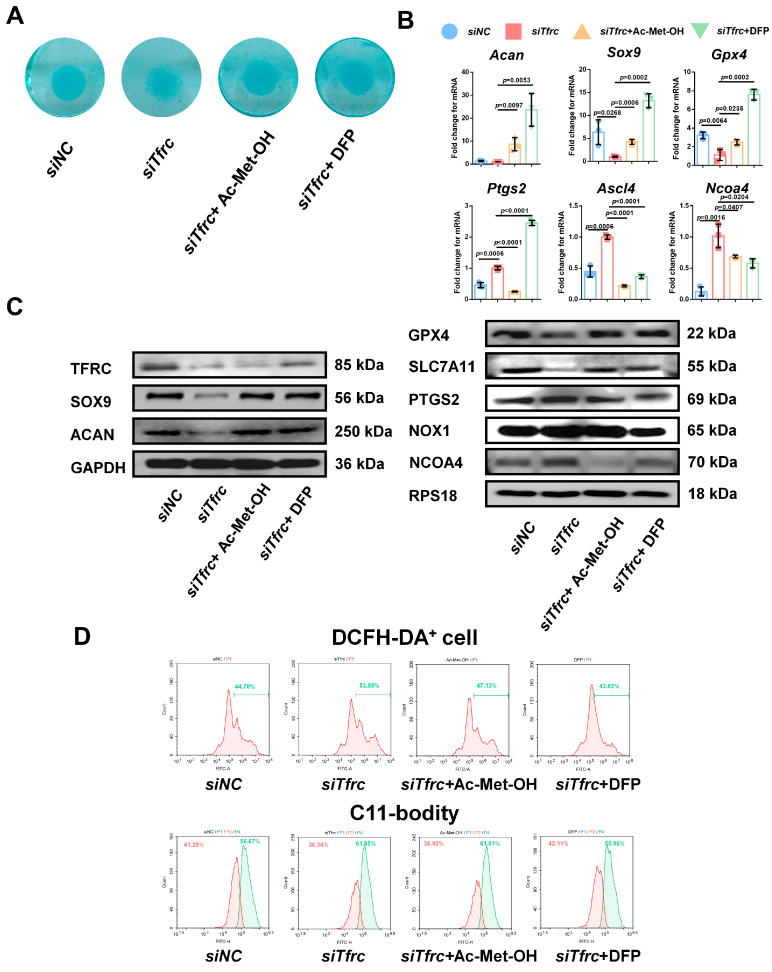
ROS scavenger Ac-Met-OH and iron chelating agent DFP restore chondrogenic differentiation of ATDC5 cells after *Tfrc* knockdown (**A**) Alcian blue staining of micromass cultivation in *siNC*, *siTfrc* and cells added with Ac-Met-OH and DFP groups. (**B**) qPCR analysis of *Sox9*, *Acan*, *Gpx4*, *Ptgs2, Ncoa4* and *Ascl4* expression in cells after treatment with Ac-Met-OH and DFP (*n* = 3, one-way ANOVA). Blue circle represents data of *siNC* group. Red square represents data of *siTfrc* group. Orange triangle represents data of *siTfrc* + Ac-Met-OH group. Green triangle represents data of *siTfrc* + DFP group. (**C**) Western blot analysis of TFRC, SOX9, ACAN, GPX4, SLC7A11, PTGS2, NOX1 and NCOA4 in *siNC*, *siTfrc*, *siTfrc +* Ac-Met-OH and *siTfrc +* DFP groups. (**D**) The proportion of DCFH-DA^+^ cells and ratio of oxidation and reduction cells detected by flow cytometry in *siNC*, *siTfrc*, *siTfrc +* Ac-Met-OH and *siTfrc +* DFP groups. Data are presented as mean ± SD.

**Figure 10 ijms-26-02724-f010:**
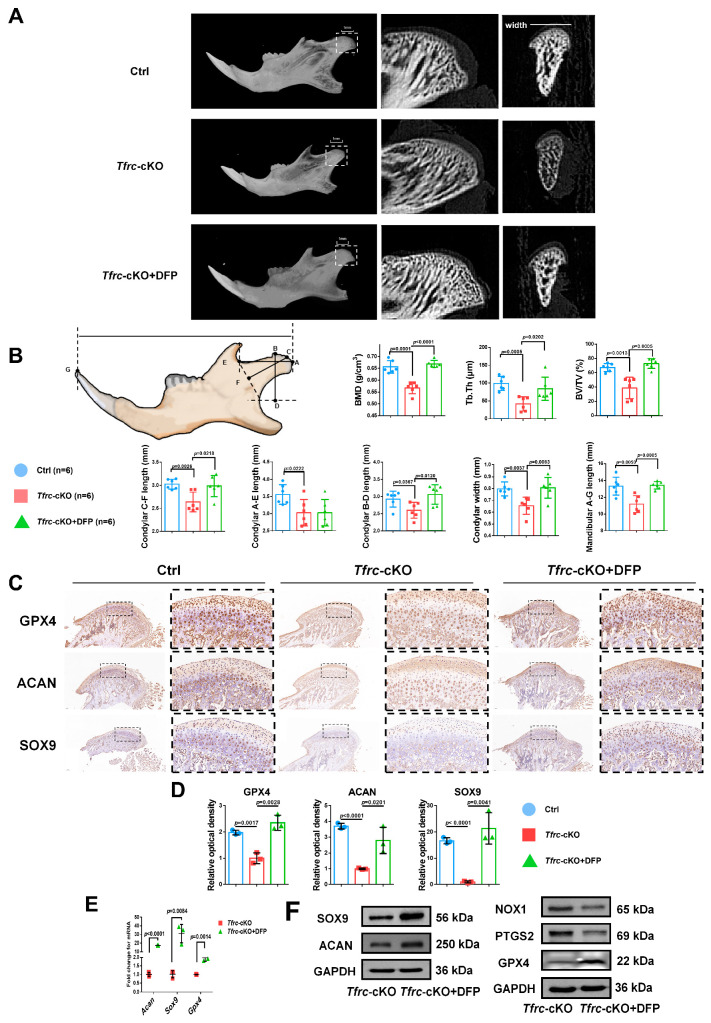
Iron chelator DFP rescues mandibular (condyle) hypoplasia of *Tfrc*-cKO mice. (**A**) The representative microCT section showing sub-cartilage bone trabecula in condyle of *Tfrc*-cKO mice and after treatment with DFP. (**B**) Bone Mineral Density (BMD), Bone Volume/Tissue Volume (BV/TV) and Trabecular Thickness (Tb.Th) reflects the basic situation of bone mass. The schematic diagram of mice mandible shows the selection and position of condylar and mandibular length measurement points (*n* = 6, one-way ANOVA). Point A: the posterior-most point of the articular surface of the condyle; Point B: the anterior-most point of the articular surface of the condyle; Point C: the outermost point of the condyle; Point D: BD was the distance from point B to the deepest point of the mandibular ramus. The foot of the perpendicular is Point D; Point E: the deepest point of the mandibular notch; Point F: CF is the distance from Point C to the line connecting Point E and the deepest point of the mandibular ramus. The foot of the perpendicular is Point F; Point G: Tip of the mandibular incisor. (**C**) IHC staining of SOX9, ACAN and GPX4 in the condyle of Ctrl, *Tfrc*-cKO and *Tfrc*-cKO mice treated with DFP. The image with a dashed border on the right is an enlarged image of the left image. (**D**) The semi-quantitative calculation results of IF and IHC (*n* = 3, one-way ANOVA). (**E**) qPCR analysis of *Sox9*, *Acan*, *Gpx4* expression in the condyle cartilage of *Tfrc*-cKO and *Tfrc*-cKO mice treated with DFP (*n* = 3, unpaired two-tailed Student’s *t* test). (**F**) Western blot analysis of SOX9, ACAN, GPX4, PTGS2 and NOX1 expression in the condyle cartilage of *Tfrc*-cKO and *Tfrc*-cKO mice treated with DFP. Blue circle represents data of Ctrl group. Red square represents data of *Tfrc*-cKO group. Green triangle represents data of *Tfrc*-cKO + DFP group. Data are presented as mean ± SD.

## Data Availability

The datasets used and/or analyzed during the current study are available from the corresponding author on reasonable request.

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
