# Peer review of "TFRC Ablation Induces Insufficient Cartilage Development Through Mitochondrial p53 Translocation-Mediated Ferroptosis"

_ijms, 2025, doi:10.3390/ijms26062724_

Round 1
Reviewer 1 Report
Comments and Suggestions for Authors
This study is highly interesting, and the authors have provided extensive data to support their hypothesis regarding the relationship between TFRC and SMH disease through ferroptosis in cartilage. However, the manuscript still needs to provide more conclusive evidence to fully validate the proposed mechanisms and strengthen the overall findings.
1. Correct the misspelling in line 20, changing "chon cartilage" to "chondral cartilage."
2. The sentence, "Overexpression and 23 knockdown of TFRC in the ATDC5 cell line were used," needs to be completed by specifying the purpose of these experiments (e.g., to investigate their role in a specific biological process or pathway).
3. The sentence, "Temporomandibular joint injection of DFP used to rescue in vivo," needs clarification—rescue from what?
4. The sentence, "TFRC was crucial for condylar cartilage development, but TFRC ablation with upregulated SLC39A14 triggers cartilage development retardation by ferroptosis," is difficult to understand. Consider rephrasing for clarity and to better explain the relationship between TFRC, SLC39A14, and ferroptosis.
5. In the introduction, the relationship between SMH disease and TFRC is not addressed. Why was TFRC chosen as the gene of interest for studying SMH?
6. In Results Section 3, the areas analyzed by IHC and IF appear different. The authors should explain why different areas were selected to confirm SLC39A14 and TFRC expression.
7. This paper lacks direct evidence linking SLC39A14 to chondrogenesis. In Results Section 4, the authors should include data on the expression of chondrogenic markers under SLC39A14 overexpression or knockdown in ATDC5 cells to strengthen the conclusions.
8. The western blotting of ACAN was not clear in the figure 8 F.
Comments on the Quality of English LanguageNeed improve
Author Response
Thank you very much for taking the time to review this manuscript. Please find the detailed responses below and the corresponding revisions in the re-submitted files. Please see the attachment.
Comments 1: Correct the misspelling in line 20, changing "chon cartilage" to "chondral cartilage."
Response 1: Thank you for pointing this out. We have corrected the misspelling here.
Comments 2: The sentence, "Overexpression and knockdown of TFRC in the ATDC5 cell line were used," needs to be completed by specifying the purpose of these experiments (e.g., to investigate their role in a specific biological process or pathway).
Response 2: Thank you for your suggestion. We have revised this sentence in methods of the abstract.
“Overexpression and knockdown of TFRC in ATDC5 cell line were used to investigate its role in a specific biological process.”
Comments 3: The sentence, "Temporomandibular joint injection of DFP used to rescue in vivo," needs clarification—rescue from what?
Response 3: Thank you for your suggestion. We have revised this sentence in methods of the abstract.
“Temporomandibular joint injection of DFP used to rescue cartilage phenotype in vivo.”
Comments 4: The sentence, "TFRC was crucial for condylar cartilage development, but TFRC ablation with upregulated SLC39A14 triggers cartilage development retardation by ferroptosis," is difficult to understand. Consider rephrasing for clarity and to better explain the relationship between TFRC, SLC39A14, and ferroptosis.
Response 4: Thank you for your suggestion. We have revised this sentence in results of the abstract.
“TFRC was crucial for condylar cartilage development. Here we reported that TFRC ablation led to condylar cartilage thickness and condyle length alteration, and induced ferroptosis of chondrocyte by upregulated SLC39A14.”
Comments 5: In the introduction, the relationship between SMH disease and TFRC is not addressed. Why was TFRC chosen as the gene of interest for studying SMH?
Response 5: we add the sentence “Considering condyle cartilage dysplasia results in SMN, we chose TFRC as our target.” in the Introduction.
Comments 6: In Results Section 3, the areas analyzed by IHC and IF appear different. The authors should explain why different areas were selected to confirm SLC39A14 and TFRC expression.
Response 6: Thank you for pointing out. We have corrected the position of the black box in IHC image of SLC39A14 of Figure 4. The areas we analyzed for IHC and IF mainly include the proliferative layer and hypertrophic layer of cartilage. The proliferative layer and hypertrophic layer serve as the main center of cartilage development. We have been aware of the limitations of two different methods in quantifying the expression of TFRC and SLC39A14. Therefore, we extracted RNA or protein from cartilage tissue to conduct qPCR or western blot for further validation.
Comments 7: This paper lacks direct evidence linking SLC39A14 to chondrogenesis. In Results Section 4, the authors should include data on the expression of chondrogenic markers under SLC39A14 overexpression or knockdown in ATDC5 cells to strengthen the conclusions.
Response 7: Thank you for your suggestion. We have changed the subheading for section 2.7 and included data on the expression of chondrogenic markers SOX9 and ACAN under SLC39A14 knockdown in ATDC5 in this section.
“To investigate the interplay between TFRC and SLC39A14 and the role of SLC39A14 on chondrogenic differentiation, we knock down SLC39A14 expression by siRNA. qPCR and western blot analysis showed decreased SLC39A14 but increased TFRC expression in siSlc39a14 group (Figure A4A & A4B). There was also increased SOX9 and ACAN expression at mRNA and protein levels in siSlc39a14 group (Figure A4A & A4B). Alcian blue staining showed chondrogenic differentiation was more active in siSlc39a14 group (Figure A4C). qPCR and western blot showed that there was increased GPX4 and decreased PTGS2, NOX1 and NCOA4 expression in siSlc39a14 group (Figure A4D). These suggested Slc39a14 ablation could up-regulate Tfrc expression thereby promoting chondrogenic differentiation through inhibiting ferroptosis.”
Comments 8: The western blotting of ACAN was not clear in the figure 8 F.
Response 8: Thank you for pointing out. We have replaced this western blotting of ACAN in Figure 8F.

Reviewer 2 Report
Comments and Suggestions for Authors
The manuscript entitled “TFRC ablation induces insufficient cartilage development through mitochondrial p53 translocation-mediated ferroptosis”. The aim of the study was to further explore how TFRC signaling regulates chon cartilage development.
Below are some suggestions:
In the Abstract:
- I suggest the authors improve the description of the methodology, it's a bit confusing to read;
- insert some morphometric data (values) in the results.
In the Introduction:
- The introduction is clear and objective, but I suggest the authors finish by including the research objective.
In the Results:
- In general, the graphs in the figures are difficult to visualize, so one suggestion is to break up the figures and divide them into plates;
- In the same way as suggested above, authors can also split up the legends, which are too extensive;
- The results presented are well described.
In the Discussion:
- I suggest starting the discussion, in the first paragraph, with a contextualization of the manuscript, objective, as well as the main results.
In the Materiasl and Methods:
-The methodology is well described, in accordance with all the procedures and analyses.
In the Conclusion:
- I suggest inserting a conclusion with final considerations, including the objective of the research, its main results according to the objective and future clinical perspectives;
- The conclusion is too limited given all the results presented.
Author Response
Thank you very much for taking the time to review this manuscript. Please find the detailed responses below and the corresponding revisions in the re-submitted files.
Comments 1: In the Abstract:
- I suggest the authors improve the description of the methodology, it's a bit confusing to read;
- insert some morphometric data (values) in the results.
Response 1: Thank you for your suggestion. We have improved the description of the methods and inserted some morphometric data in the results in the abstract.
“Methods: TFRC, SLC39A14, chondrogenic markers and ferroptosis-related signals were detected in condylar cartilage of postnatal mice at different time points and Tfrc cartilage conditional knockout (Tfrc-cKO) mice through immunofluorescence, immunohistochemical staining, qPCR assays. Overexpression and knockdown of TFRC in ATDC5 cell line were used to investigate its role in a specific biological process. Co-immunoprecipitation used to verify protein-protein interaction in vitro. Ferroptosis inhibitor Fer1, Ac-Met-OH and DFP were used for rescue assay in vitro. Temporomandibular joint injection of DFP used to rescue cartilage phenotype in vivo.”
“Results: TFRC was crucial for condylar cartilage development. Here we reported that TFRC ablation led to condylar cartilage thickness and condyle length alteration, and induced ferroptosis of chondrocyte by upregulated SLC39A14. Mitochondrial p53 translocation was involved in TFRC-SLC39A14 switch by SLC39A14 ubiquitination degradation. Fer1, Ac-Met-OH and DFP inhibited ferroptosis and restored chondrogenic differentiation in vivo. Temporomandibular joint injection of DFP could rescue the cartilage phenotype.”
Comments 2:In the Introduction:
- The introduction is clear and objective, but I suggest the authors finish by including the research objective.
Response 2: Thank you for your suggestion. We have included the research objective in the last paragraph of the introduction section.
“In this study, we aimed to explore the role of TFRC in condylar cartilage development and elucidate the mechanism of TFRC-SLC39A14 switch in cartilage hypoplasia. Here we demonstrated that the stable expression of TFRC was a crucial for the fate of articular cartilage or upregulation of SLC39A14 with excessing non-transferrin bound iron (NTBI) transportation would induce ferroptosis. The dominant expression switch from TFRC to SLC39A14 in chondrocytes was mediated by mitochondrial p53 translocation-induced SLC39A14 ubiquitination degradation. Our findings have deepened the understanding of the roles played by TFRC and SLC39A14 in cartilage development, as well as the regulatory interplay between TFRC and SLC39A14, offering new perspectives for etiology, pathogenesis and therapy of SMH.”
Comments 3: In the Results:
- In general, the graphs in the figures are difficult to visualize, so one suggestion is to break up the figures and divide them into plates;
- In the same way as suggested above, authors can also split up the legends, which are too extensive;
- The results presented are well described.
Response 3: Thank you for your suggestion. We have divided original Figure 3 into Figure 3 and Figure 4, original Figure 7 into Figure 9 and Figure 10 to increase readability.
Corresponding subsection, serial numbers and Figure legends have been modified in the manuscript.
Comments 4: In the Discussion:
- I suggest starting the discussion, in the first paragraph, with a contextualization of the manuscript, objective, as well as the main results.
Response 4: Thank you for your suggestion. We have included the objective of our study in the first paragraph in the discussion.
“In this study, we intended to explore the role of TFRC on condylar cartilage development and the underlying mechanism of TFRC-SLC39A14 switch on cartilage hypoplasia. Here, our study reveals that sTFRC reaches its peak during the pubertal stage of cartilage development. TFRC, which binds to transferrin and facilitates its transfer, could promote the development of condyle cartilage. TFRC ablation would trigger a compensatory upregulation of SLC39A14, facilitating the transfer of NTBI into the cytoplasm. This switch in expression leads to ferroptosis in chondrocytes, a process in which mitochondrial p53 translocation could mediate SLC39A14 ubiquitination degradation (Graphical Abstract).”
Comments 5: In the Materials and Methods:
-The methodology is well described, in accordance with all the procedures and analyses.
Response 5: Thank you for your evaluation.
Comments 6: In the Conclusion:
- I suggest inserting a conclusion with final considerations, including the objective of the research, its main results according to the objective and future clinical perspectives;
- The conclusion is too limited given all the results presented.
Response 6: Thank you for your suggestion. We have included the objective and future clinical perspective in the conclusion.
“In this study, we aim to delve into the role of TFRC on condylar cartilage development and the mechanism of TFRC-SLC39A14 switch on cartilage hypoplasia. We have determined that TFRC expression peaks at the puberty of mandible growth and TFRC ablation leads to inadequate cartilage development by upregulation of SLC39A14. The switch in TFRC-SLC39A14 expression initiates ferroptosis of chondrocyte, characterized by NTBI accumulation, lipid peroxidation, and antioxidant mechanism disruption. Decreased ubiquitination degradation of SLC39A14 mediated by mitochondrial p53 translocation lead to TFRC-SLC39A14 switch. Besides, iron chelating agent DFP could rescue cartilage phenotype by inhibiting ferroptosis. Our study not only unveils a novel mechanism, but also offers new perspectives on the etiology, pathogenesis, and treatment of skeletal mandibular hypoplasia.”
